# Electrochromic Performance and Capacitor Performance of α-MoO₃ Nanorods Fabricated by a One-Step Procedure

**Ying Duan ¹, Chen Wang ¹, Jian Hao ², Yang Jiao ³, Yanchao Xu ³ and Jing Wang ¹,***

¹   School of Light Industry, Harbin University of Commerce, Harbin 150028, China; hsdduany@163.com (Y.D.); 13844636005@163.com (C.W.)
²   State Key Laboratory of High-Efficiency Utilization of Coal and Green Chemical Engineering, Ningxia University, Yinchuan 750021, China; haojian@nxu.edu.cn
³   College of Geography and Environmental Sciences, Zhejiang Normal University, Jinhua 321004, China; yangjiao@zjnu.edu.cn (Y.J.); xu8yanchao@163.com (Y.X.)
*   Correspondence: wangwangmayong@126.com; Tel.: +86-451-8486-5185

**Abstract:** In this paper, we propose for the first time the synthesis of α-MoO₃ nanorods in a one-step procedure at mild temperatures. By changing the growth parameters, the microstructure and controllable morphology of the resulting products can be customized. The average diameter of the as-prepared nanorods is about 200 nm. The electrochromic and capacitance properties of the synthesized products were studied. The results show that the electrochromic properties of α-MoO₃ nanorods at 550 nm have 67% high transmission contrast, good cycle stability and fast response time. The MoO₃ nanorods also exhibit a stable supercapacitor performance with 98.5% capacitance retention after 10,000 cycles. Although current density varies sequentially, the nanostructure always exhibits a stable capacitor to maintain 100%. These results indicate the as-prepared MoO₃ nanorods may be good candidates for applications in electrochromic devices and supercapacitors.

**Keywords:** α-MoO₃ nanorods; controlled synthesis; one-pot water bath; electrochromic properties; capacitor performance

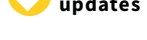



## 1. Introduction

In the last few years, the problems of global warming and energy shortage have become more and more serious problems. Curbing greenhouse gas emissions and developing clean alternative energy sources and advanced management and energy storage system have become the top priorities. Electrochromism and electrochemical capacitors based on layered transition metal oxides nanostructures have attracted the attention of many technologists and researchers for potential applications for energy storage because of their ability to insert the ions to a wide range of locations [1–3]. Among many transition metal oxides, Molybdenum trioxide (MoO₃) has become a promising candidate material for energy storage applications due to its relatively low pollution to the environment, layered nanostructures with a variety of oxidation states, high-energy density and inexpensivity [4–6].

MoO₃ has a wide range of applications in many fields such as electrochromics [7–9], photochromics [10,11], electrochemical capacitors [12,13], electrocatalytic activities [14,15], gas sensors [16–19], and lithium-ion batteries [20–24], which makes it a more popular transition metal oxide. Recently, the electrochromic properties of MoO₃ nanomaterials have attracted much attention. It is one of the best candidates for the electrically induced discoloration device manufacturing materials. It has a fast response, high coloration efficiency, low activation potential gradient and color change. Compared with the electrochromic performance of WO₃, the color state of MoO₃ absorbs light stronger and more uniformly and has better opening memory performance. Lei et al. have prepared MoO₃ nanobelts and studied their photochromic and electrically induced discoloration performance [25].

Cai et al. have reported on the morphology-controlled flame synthesis of $MoO_3$ nanobelt arrays [26]. In addition, the material of $MoO_3$ as energy storage devices, especially the electrochemical capacitor, due to its high power density, energy density and specific surface area, in the future of energy storage applications received great attentions. Shakir et al. studied tin-oxide-coated molybdenum oxide nanowires for use in high-performance supercapacitor equipment [27]. Furthermore, Aravinda created composite electrodes of $MoO_3$/carbon nanotubes for an electrochemical supercapacitor [28]. Previously, the majority of research has been centered on the functions of the thermodynamically steady section of orthorhombic $MoO_3$ materials ($\alpha$-$MoO_3$). Its zigzag chains and special layers shape current open channels for ion intercalation or diffusion [29–31]. This structural property makes $\alpha$-$MoO_3$ nanomaterials promising materials for electrochromism and capacitor functions. Many methods are used to prepare $MoO_3$ nanomaterials [28–31] for electrochemical performance. Among these procedures, the water bath method is better for forming the stable films and easier to synthesize the materials.

In this paper, we employed a facile one-pot water bath route to synthesize controlled rod-like $\alpha$-$MoO_3$ nanomaterials. We record for the first time that controlled one-pot water bath synthesized rod-like $\alpha$-$MoO_3$ nanomaterials used for electrochromism and capacitor applications. The organized $\alpha$-$MoO_3$ nanorods exhibit excessive transmittance modulation 67%, excellent coloring efficiency, and moderate color alterations. In addition, the $\alpha$-$MoO_3$ nanorods also exhibit good cycling stability for supercapacitors. It is anticipated that such $MoO_3$ nanorods produced using a one-pot water bath technique are predicted to have future uses in smart windows and energy storage devices.

## 2. Section of Experimentation

The materials used in this experiment are $Na_2MoO_4 \cdot 2H_2O$ and $HNO_3$. The producer of $Na_2MoO_4 \cdot 2H_2O$ is Shanghai Chaoyan Biotechnology Co., Ltd. (Shanghai, China), and the producer of $HNO_3$ is Xinxiang Haolong Chemical Co., Ltd. (Xinxiang, China).

The usual synthesis process for $MoO_3$ nanorods is as follows. To begin, 3 mmol of $Na_2MoO_4 \cdot 2H_2O$ were mixed into 40 mL of distilled water and stirred (magnetic stirring). Then, while vigorously swirling, commercial $HNO_3$ was progressively added to the $Na_2MoO_4$ solution until the pH was 4–6. Following that, the combined solution was placed into a 50 mL beaker after being stirred for 2 h. Then the above mixture was taken into a water bath with an indium tin oxide (ITO) followed by adjusting the reaction temperature 170 °C for 15 h. The solution was then cooled to room temperature in a still condition. Sediment washing with deionized water and ethanol several times. Finally, the product was dried in a 50 °C chamber for 12 h.

In addition, we have prepared the other experimental conditions for comparisons. With the reaction time consistent, we researched the reaction temperatures for 90 °C and 240 °C. We also studied the reaction times for 5 h and 20 h, respectively. These experiments for comparisons, the dried temperature and the time are the same as the above conditions.

The microstructure and the micromorphology of the as-prepared samples were characterized by powder X-ray diffraction (XRD, D/max-rB 12 kW, Bo Yue Instruments (Shanghai) Co., Ltd., Shanghai, China), Raman Spectroscopy (HR800, Shanghai Ruhai Optoelectric Technology Co., Ltd., Shanghai, China), a high-resolution scanning electron microscope (HR-SEM, S-4800, Hitachi, Wuxi Nuohe Ruite Automation Equipment Co., Ltd., Wuxi, China) operated at 20 kV and equipped with energy-dispersive X-ray spectroscopy (EDX, Shenzhen Tianwei Instrument Co., Ltd., Shenzhen, China), and transmission electron microscopy (TEM, Tecnai G2 F30, Zhengzhou Shuren Technology Development Co., Ltd., Zhengzhou, China). Transmission spectra were collected from a UV-vis-NIR fiber optic spectrometer (Ocean Maya 2000-Pro, Chengdu Guangchi Technology Co., Ltd., Chengdu, China). The incident light was perpendicular to the sample surface. All electrochemical measurements were carried out on an electrochemical workstation (CHI660E, Shanghai Chenhua Device Company, Shanghai, China).

The MoO$_3$ nanorods were prepared by Lei et al., reported the molybdenum trioxide nanorod coating film for electrochromic measurement. The electrochromic properties of MoO$_3$ nanorods were investigated by using a 1 M lithium perchlorate (LiClO$_4$) three-electrode electrochemical cell in a propylene carbonate (PC) electrolyte. In all experiments, cyclic voltammetry (CV) and regular voltage traits have been recorded with an electrochemical station (CHI 660E). The working electrode was a MoO$_3$ nanorod film that was vertically placed into the electrolyte, the counter electrode was a Pt film, and the reference electrode was a saturated calomel electrode (SCE). CV curves were measured with a sweep rate of 0.05 V/s between 1.0 and −1.0 V.

The MoO$_3$ nanorods used for the capacitor measurements was prepared as follows. Electrochemical measurements were made using a typical three-electrode test cell on the CHI660E electrochemical workstation. All measurements were made in an aqueous solution of 0.5 M Na$_2$SO$_4$ electrolyte at room temperature. Working electrodes for supercapacitors were created by mixing 80 wt.% of active materials with 10 wt.% of carbon black and 10 wt.% of the binder PVDF (polyvinylidenefluoride). To create a homogenous mixture, a tiny quantity of NMP (*N*-methylpyrrolidone) was added to the composites. To make a working electrode, the resultant mixture was deposited onto a 1 cm$^2$ Ni-foam current collector. The electrode was then dried at 80 °C for 12 h. The counter and reference electrodes were a Pt film and a saturated calomel electrode (SCE), respectively. Cyclic voltammograms were obtained at various scan speeds ranging from 5–80 mV/s within a potential range of −0.2 to 0.6 V. Charge-discharge cycle performance experiments were carried out at various current densities and potentials ranging from −0.2 to 0.6 V.

## 3. Results and Discussion

The XRD pattern of the obtained products is shown in Figure 1a. All diffraction peaks conform to the standard $\alpha$-MoO$_3$ pattern (JCPDS No. 65-2421). No peak was detected in other phases, indicating high purity of the product. Sharp peaks show that the preparation of the product is highly crystallized. The Raman spectrum of the obtained product at room temperature is shown in Figure 1b. The main Raman lines at 666, 820 and 995 cm$^{-1}$ are related to the stretching vibrations and can be assigned to the $\alpha$-MoO$_3$ phase. The Raman peaks at 666 cm$^{-1}$ and 820 cm$^{-1}$ might be due to the Mo–O–Mo bonds of monoclinic MoO$_3$ [32]. The band about 995 cm$^{-1}$ is thought to be representative of Mo=O terminal bonds. The band at 285 cm$^{-1}$, on the other hand, is ascribed to crystal lattice vibration [33]. SEM was used to characterize the morphology. The low magnification SEM image Figure 2a shows large quantities of rod-like nanomaterial. As seen in Figure 2a, these nanowires are evenly dispersed and have a width of around 200 nm. Using the hole portion of Figure 2a for SEM mapping experiments, the results indicate the presence of Mo and O elements.

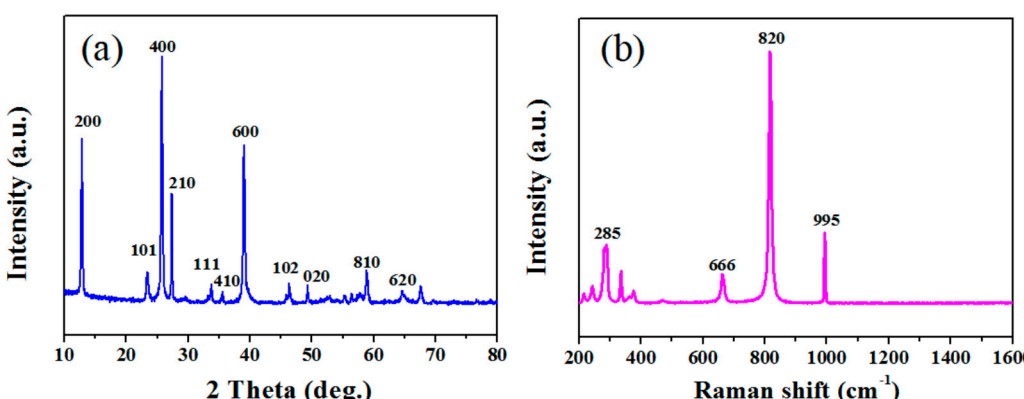

**Figure 1.** (**a**) XRD pattern and (**b**) Raman spectrum of the synthesized products.

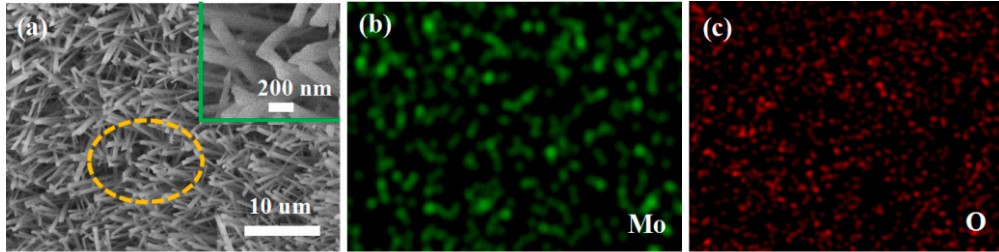

**Figure 2.** Morphology of the as-synthesized MoO3 nanorods (**a**) Low magnification SEM image and the inset is the high magnification SEM images; (**b**,**c**) SEM mapping images of the yellow section from (**a**).

A series of controlled experiments was conducted to find the best reaction conditions. First, keeping the other growth parameters constant, only the reaction time was changed (as shown in Figure 3a). After five hours, $MoO_3$ nanorods with nanoparticles on the surface were detected. When the response time increased to 20 h, the sizes of the nanorods grew larger, as shown in Figure 3c. Temperature also has an essential influence on controlling the morphology of the obtained material. By keeping the other growth parameters constant, only the temperature was changed. At 90 °C, most of the material is nanoparticles as shown in Figure 3d. Increasing the temperature to 240 °C, the nanorods are broken or aggregated together (as shown in Figure 3f). The above results indicate that controlling the growth parameters is essential to obtain the desirable nanostructures. The above results indicate that the best reaction time is 15 h and the best reaction temperature is 170 °C (as shown in Figure 3b,e). The following is a putative growth process for $MoO_3$ nanorods.

$$Na_2MoO_4 \cdot 2H_2O \rightarrow 2Na^+ + MoO_4{}^{2-} + 2H_2O \tag{1}$$

$$HNO_3 \rightarrow H^+ + NO^{3-} \tag{2}$$

$$xH^+ + MoO_4{}^{2-} \rightarrow H_xMoO_4 \tag{3}$$

$$H_xMoO_4 \rightarrow H_2O + MoO_3 \tag{4}$$

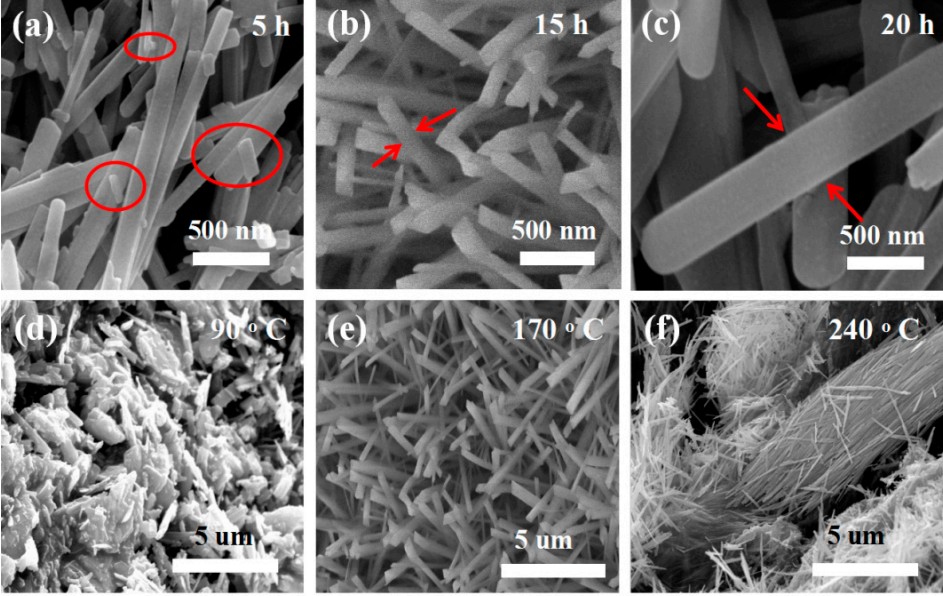

**Figure 3.** Growth control of the product morphology. (**a–c**) SEM images of the product with the reaction time 5 h, 15 h and 20 h at the same reaction temperature 170 °C; (**d–f**) SEM images of the product with the reaction temperature 90 °C, 170 °C and 240 °C at the same reaction time 15 h, respectively.

In the initial stage, $Na_2MoO_4 \cdot 2H_2O$ and $HNO_3$ are ionized to create $Na^+$, $MoO_4^{2-}$, $NO^{3-}$, and $H^+$. After that, $H^+$ reacts with $MoO_4^{2-}$ to form $H_xMoO_4$. $H_xMoO_4$ will be converted into tiny $MoO_3$ nucleation particles under hydrothermal conditions. Under hydrothermal conditions, $H_xMoO_4$ reacts to form $MoO_3$ nucleating particles. Because of their high surface energy, they will attract and gather with each other. When the nearby nanoparticles begin to grow, their surface energy will shrink rapidly [34,35]. In the process of continuous crystal growth, the nanoparticles in each aggregate will determine their own growth direction and further grow as nucleation sites [36]. Similarly, when the neighboring nanoparticles gather together, the surface energy decreases rapidly [37]. At the reaction time 5 h, these nanoparticles aggregate to form nanorods and these nanorods with the different sizes, as shown Figure 3a. Additionally, we increased the time to 15 h. These particles become nanowires, as shown in Figure 3b. Further taking the reaction time for 20 h, the nanorods become bigger with the diameter about 500 nm as shown in Figure 3c. When adjusting the temperature with other conditions consistent, firstly taking the reaction temperature 90 °C, these nanoparticles formed some blocks and aggregated together as shown in Figure 3d. When the temperature is adjusted to 170 °C, some nanorods can be found as shown in Figure 3e. Further increase the time to 20 h and increase the temperature to 240 °C, the resulting nanowires will become larger, aggregated or fragmented. The rod-like $MoO_3$ structure is created in two steps: nucleation and growth. These growth processes are determined by the anisotropic structure and reaction conditions. Therefore, the growth mode of $MoO_3$ crystal can determine the morphology of $MoO_3$ crystal grown into nanorods [38,39].

In order to understand the electrochromic properties of $MoO_3$ rods with different shapes more accurately, we observed the transmittance and response time of $MoO_3$ particles and $MoO_3$ nanorod clusters under the same conditions. Figure 4 shows the transmittance and response time changes at the equal scan rate 50 mV/s. The results demonstrate that the transmittance changes are $MoO_3$ rods (47%) > $MoO_3$ particles (36%) > $MoO_3$ nanorods cluster (22%). The colored and bleached response time are $MoO_3$ nanorods (11 s, 20 s) < $MoO_3$ particles (15 s, 30 s) < $MoO_3$ nanorods cluster (20 s, 35 s). The results show that the transmittance of $MoO_3$ nanorods changes the most and the response time is the fastest.

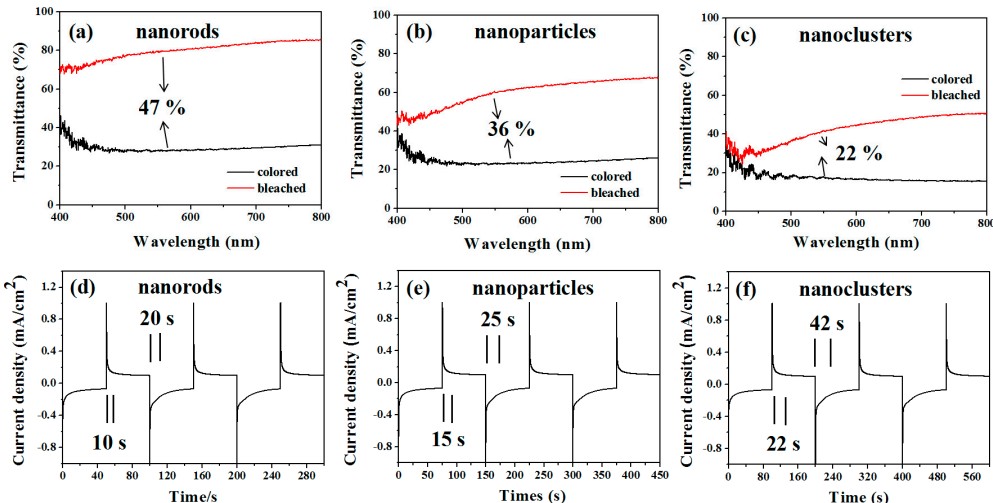

**Figure 4.** Cyclic voltammograms of $MoO_3$ nanostructured electrodes at a scanning rate of 50 mV/s. The transmittance changes of (**a**) nanorods, (**b**) nanoparticles, and (**c**) nanoclusters. Response time changes of (**d**) nanorods, (**e**) nanoparticles, and (**f**) nanoclusters.

The absorbable light sensitivity of $MoO_3$ is 400–800 nm, which can be widely used in electrochromic materials. The transmission spectra of the prepared $MoO_3$ nanorod films under different voltages (−1–+1 V) are shown in Figure 5. The transmission spectra of $MoO_3$ nanorod films in the range of 550–800 nm show the widest variation of $MoO_3$ as shown

in Figure 5a with the scan rate 10 mV/s. At 550 nm, the transmittance integral change of colored and bleached states is 67%. In the voltage range (+1 to −1 V), the color change occurs only in negative voltage and dark blue. After moving to the positive voltage, the film was bleached and returned to the original transparent state. The different color states are as shown in the inset of Figure 5a. Electrochromic response and switching time can greatly affect the choice of electrochromic materials. The current-time response is depicted in Figure 5b. The dyeing reaction time and bleaching reaction time are about 2.4 s and 3.2 s respectively. Scientific research shows that the electrochromic mechanism of $MoO_3$ in lithium-ion electrolytes can be reasonably explained as follows: the insertion/delamination of electrons and the charge balance of lithium ions determine the coloring/bleaching process. Furthermore, coloration efficiency (CE) is one of the most often used methods for determining the effective change in optical transmission with applied voltage. When a film is tinted (Tc) or bleached (Tb) at a certain wavelength, CE is the ratio of optical density (OD) to injected (or expelled) charge density (q) per unit area. Q can be calculated from the integral current time response curve. The optical density change is calculated as follows:

$$\eta(\lambda) = \Delta OD\ (\lambda)/Q; \Delta OD(\lambda) = \log(T_b/T_c)$$

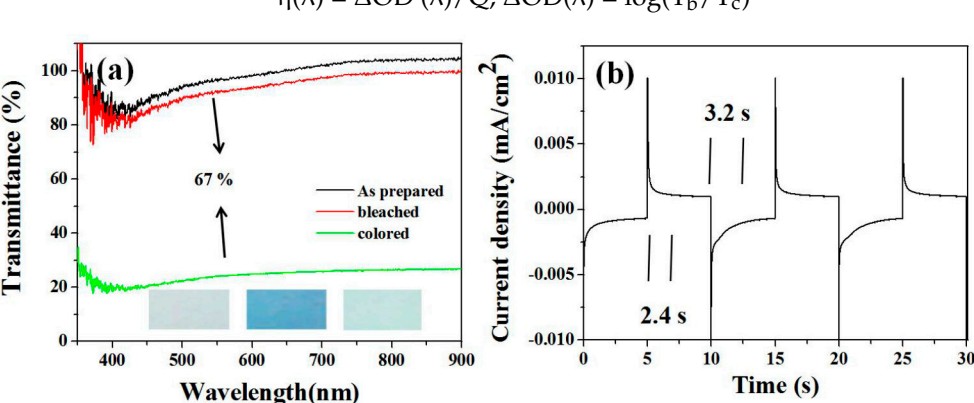

**Figure 5.** Cyclic voltammograms of $MoO_3$ nanorod electrodes at a scanning rate of 10 mV/s (**a**) in the potential range of −2 to +2 V and the optical transmittance spectra of $MoO_3$ nanorods in the colored and bleached states. The inset images are the photographs of as-prepared $MoO_3$ and the $MoO_3$ nanorod film in the colored and bleached states. (**b**) Chronoamperometric curve of the $MoO_3$ nanorods film.

The coloration efficiency ($\eta$) is 46.5 cm$^2$ C$^{-1}$ at 550 nm for the $MoO_3$ nanowires.

In order to explore whether the reaction products have potential application value in supercapacitor electrodes, we examined the electrochemical performance with $Na_2SO_4$ as the electrolyte. We compared the $MoO_3$ nanorods capacitors' performance with $MoO_3$ particles and $MoO_3$ nanorod clusters. Cyclic voltammograms at the same scan rates 5 V/s with the potential window from −0.2 to +0.6 V and the same current density 2.5 A/g are shown in Figure 6a,b. It also shows that the capacitors are $MoO_3$ rods (451 F/g) > $MoO_3$ particles (223 F/g) > $MoO_3$ cluster (107 F/g).

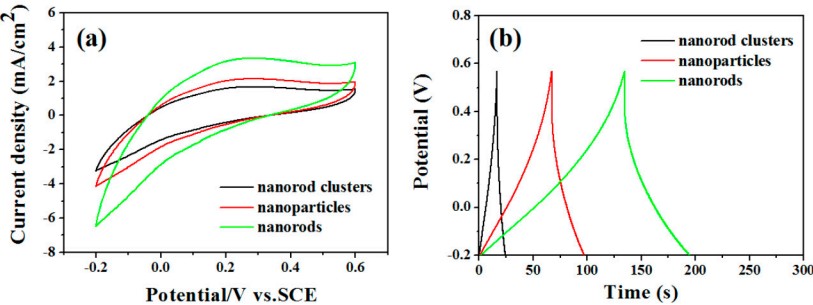

**Figure 6.** Electrochemical characterization of the $MoO_3$ nanowires, nanoparticles, and nanorod clusters. (**a**) Cyclic voltammograms at the same scan rates 5 V/s with the potential window from −0.2 to +0.6 V and (**b**) Charge/discharge curves at the same current density 2 A/g.

Figure 7a shows cyclic voltammograms at various scan speeds ranging from 5 to 80 mV/s with the potential window ranging from −0.2 to +0.6 V. The peak current increases with the increase of scanning rate, which indicates that $MoO_3$ structure has the function of charge transfer and ion diffusion. More scientifically, a pair of indistinct redox peaks can be observed in the potential range of −0.2 to +0.6 V at all scanning rates. Compared with the electric double-layer capacitor, the pseudo-capacitance behavior of electrochemical adsorption redox reaction at the interface of electrode and electrolyte is obvious [40]. The correspondence to the conversion between different valence states is as follows:

$$MoO_3 + xNa^+ + xe^- \leftrightarrow Na_xMO_3$$

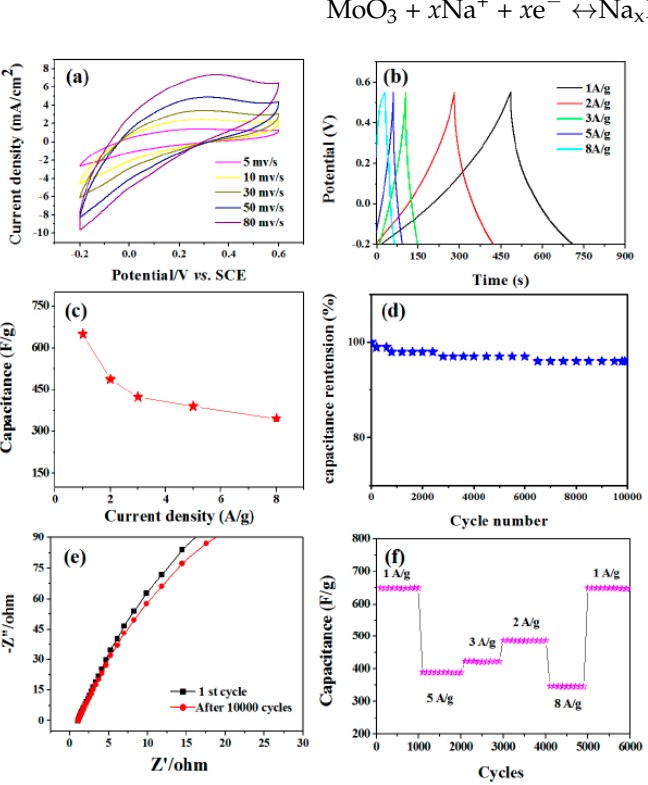

**Figure 7.** Electrochemical characterization of the $MoO_3$ nanorod film in a three electrode system. (**a**) CV curves of $MoO_3$ nanorods at a scan rate of 5, 10, 30, 50, and 80 mV/s. (**b**) Charge/discharge curves at current densities of 1, 2, 3, 5, and 8 A/g. (**c**) Current density dependence of the specific capacitance. The inset is the cycling property with the density of 3 A/g. (**d**) Cycling performance under the condition of different current densities. (**e**) The electrical conductivity of the product for the first and last 10,000 cycles of the Nyquist plots. (**f**) The cycling performance of the $MoO_3$ nanorods tested at different current densities.

Galvanostatic charge-discharge curves were performed at different current densities of 1, 2, 3, 5, 8 A/g as shown in Figure 7b. Specific capacitances were calculated according to $Csp = I \times \Delta t/(m \times \Delta V)$ where I is the constant discharge current, $\Delta t$ is the discharge time, $\Delta V$ is the potential drop during the discharge time, and m is the mass of the $MoO_3$ nanorods in the electrode. The calculated capacitance as a function of the discharge current density is plotted in Figure 7c. The $MoO_3$ nanorod electrode exhibits a capacitance of 650, 487.5, 423.5, 389.8, and 345.6 F/g at current densities of 1, 2, 3, 5, and 8 A/g, respectively. By evaluating the specific capacitance with respect to the charge-discharge cycle number, the electrochemical stability of $MoO_3$ nanorods was investigated. For 1000 cycles, the cycling procedure was carried out at a current density of 3 A/g. Figure 7d depicts the change in the *Csp* value of $MoO_3$ nanorod electrodes as a function of cycle number. After 10,000 cycles, the capacitance retention is 98.5 percent. We then investigated the electrical conductivity of the product for the first and last 10,000 cycles, as indicated by the Nyquist plots in Figure 7e. There are no obvious changes in impedance diagram. This further confirms that the as-prepared products have good cycle stability. Figure 7f shows the cycle performance as a function of density. It exhibits a stable capacitance of 650 F/g in the first 100 cycles and a charge/discharge density of 1 A/g. Further studies show that the nanostructures exhibit stable capacitance and do not change with the order of current density. When the current density returns to 1 A/g, the capacitance (650 F/g) is almost unchanged. The experimental results show that the prepared materials have high electrochemical stability and can be used in capacitors for a long time.

From the above results, we can explicitly conclude that $MoO_3$ nanorods have an advantage over others, such as electrochromic and supercapacitor. However, the arrangement among the $MoO_3$ nanorods is relatively loose, and the ions are easy to diffuse inside. In addition, the unique microstructure composed of one-dimensional nanorods enhances the kinetics of ion and electron transport inside the electrodes materials and the electrode–electrolyte interface [41]. Finally, the materials are nanoscale. Compared with nanoparticles and nanorod clusters, the surface area of nanorods is larger, which provides more reactive areas and improves the utilization of active substances [42,43]. Therefore, nanorods have better electrochromic performance than nanoparticles and nanorods clusters in this experiment.

## 4. Conclusions

In summary, α-$MoO_3$ nanorods were synthesized by a simple controlled hydrothermal reaction under mild conditions. The average radius of the synthesized nanorods is about 100 nm. According to the experimental results, the possible mechanism of $MoO_3$ nanorods growth was proposed. Electrically induced discoloration characteristic of synthetic product was studied, as well as the capacitor performance. The electrochromic result shows that the as-prepared nanorods have better transmission modulation and coloring efficiency. Additionally, the products exhibit a stable supercapacitor performance with 95% capacitance retention after 10,000 cycles. These $MoO_3$ nanorods are expected to be used in electrochromic devices and large-scale energy storage devices.

**Author Contributions:** Conceptualization, Y.D. and J.W.; methodology, C.W.; software, J.H.; validation, J.W.; formal analysis, J.H.; investigation, Y.J.; data curation, Y.X.; writing—original draft preparation, Y.D. and J.W.; writing—review and editing, J.W.; funding acquisition, J.W. All authors have read and agreed to the published version of the manuscript.

**Funding:** We thank the young scientist foundation of Harbin University of Commerce (No. 18XN051), Scientific research project of Harbin University of Commerce (No. 2019DS094) and the National Natural Science Foundation of China (No. 52002099).

**Institutional Review Board Statement:** Not applicable.

**Informed Consent Statement:** Not applicable.

**Conflicts of Interest:** The authors declare no conflict of interest.

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
