# Peer review of "Electrochromic Performance and Capacitor Performance of α-MoO3 Nanorods Fabricated by a One-Step Procedure"

_coatings, doi:10.3390/coatings11070783_

Round 1

Reviewer 1 Report

The paper by Wang and coworkers presents interesting results on the production of MoO3 nanorods and tests of their electrochromic and capacitor performance.  I am not convinced that Coatings is necessarily the best choice of journal as the focus is on the production and applications of a nanomaterial.  Possibly a different MDPI journal such as Nanomaterials is more appropriate.  That aside, there are a number of issues that should be dealt with prior to publication. 

  1. The Introduction does not provide any details on how the present one-step procedure is better than previous methods for production of MoO3 Brief details on previous methods should be added.  In this respect please consider updating the title, possibly to “by a one-step procedure”, rather than “by one-step controlled water bath”.  Alternately, “by a one-step controlled temperature procedure”.    

  2. The instrument manufacturer is missing for some cases in the paragraph starting on line 75.

  3. Line 84 notes that ref 11 by Zheng et al provides the method for film preparation but ref 11 (line 303) has different authors. Please correct references/citations.  Also provide a brief description of the method used to prepare films and confirm if the same film type was used for all experiments.

  4. Line 118 notes that the nanorods have a width of approximately 200 nm in Fig 2b. There is no scale bar in Fig 2b or 2c which makes it a bit hard to assess this.  However, the inset in Figure 2a shows nanorods with a width of much less than 200 nm, probably close to 50 nm.  Figure 2b and c should have scale bars.  Furthermore, if the width is of significant interest here, then more convincing analysis should be presented.  Since Figure 3 presents data for nanorods prepared with variable reaction times and temperatures, the conditions used to produce the sample in Figure 2 should be noted in the caption for ease of comparison. 

  5. Lines 129-131 note that small nanoparticles are detected on the surface of the nanorods for the 5 hr reaction (Fig 3a). It appears that many of the smaller features are actually short nanorods, not nanoparticles which should have all 3 dimensions less than 100 nm (ISO definition). 

  6. Figure 3 should provide the temperature for parts a-c and the time for parts d-f.

  7. The conclusions based on the SEM images are very qualitative and there appear to be multiple sizes of individual nanorods and clusters in most images (Figures 2 and 3). If the goal is to correlate particle morphology with the behavior of the various samples in electrochromic or capacitor applications, then it would be important to provide quantitative data on the different morphologies, rather than the very qualitative conclusions presented.  This is important for Figure 4 where the plots are labelled as nanorods, nanoparticles and nanoclusters, a conclusion this is not well-justified by the data shown in Figure 3.   

  8. The transmittance plots in Figure 4a-c are labelled as cyclic voltammograms. I assume they are transmittance results measured concurrently with CV? 

  9. The sentence in lines 180-181 is confusing. I assume the authors mean that the human eye is sensitive to light in the range of 400-800 nm and this is a useful range for MoO3 ?

  10. It is not clear what the statement on line 267 of the Conclusions is based on. It is stated in line 136-137 that the optimal conditions for are 170 C and 12 hr.  Assuming this correspond to Fig 2b (although this is not stated in the figure caption), it does not look like the nanorods have a radius of 100 nm. 

  11. The English grammar and usage are rather poor in many places. The paper should be edited by a native English speaking person prior to publication. 

Reviewer 2 Report

This article shows some interesting results. Authors reported a design with controlled wet methods to fabricate three kinds of MoO3 in forms of nanorod, nanoparticle and nanorod cluster. Particularly, the interested α-MoO3 nanorod showed with good physical properties and electrical properties such as transmittance modulation and capacitance retention respectively compared to other two MoO3. Potentially, this interested α-MoO3 nanorod will be a good candidate for the electrochromic devices and superconductance applications.

  1. Suggest the experimental section should add portion of a series of controlled experiments (lines 127-138) to make explicit in the fabrication of three kinds of MoO3 in experimental section due to the importance of different processing time.
  2. In experiment section (line71), ..170 °C for 15 h.. is different from statement in the text, for example 170 °C for 12 h (line 137, line 159) to process MoO3 nanorod. Need to have consistency throughout the article.
  3. In Fig, 3 (e), the label in the right hand corner of the box should be 170 °C not 120 °C , in Fig. 3 of lines 146-148, missing the captions of (d), (e), and (f) and their illustrations.
  4. In Fig. 7 of line 232, missing (e) and (f) captions and their illustrations.
  5. In line 197, TC should be Tc and TB should be Tb.
  6. In line 245, need to elaborate more in the text on physical impedance Z meaning of Nyquist plot in Fig. 7e (missing).
  7. In the ref. 38, the paper title, the name of Journal (pages) and Doi did match each other.

Reviewer 3 Report

The authors present a study on MoO3 nanorod synthesis and the investigation of electrochromic properties and capacitor performance in corresponding films.

Parts of the paper are poorly edited and it does not only concern the English language. Without discussing the science, some of the problems are given below.

  1. In line 83 it says that the film "was prepared by Zheng et al11 had reported". I guess it means that the film was prepared following a procedure published by Zheng et al., unfortunately there is no author "Zheng" involved in reference 11.
  2. In figure 3 the morphology of several MoO3 nanorod samples is shown with SEM images. For the time series 5 h, 12 h and 20 h, the corresponding temperature is not mentioned. For the temperature series 90°C, 170°C and 240°C the time is not given. For fig. 3b the temperature should be "170°C" instead of "120°C" according to the main text. The scale bars should be marked with "5 µm" instead of "5 um". The red circles and arrows given in the figure have to be explained in the figure caption and in the text. In addition to all that there are the typos "of. the", "of.the", "oC" in the figure caption.
  3. In figure 4c and 4f it should say "nanoclusters" instead of "nanoculsters".
  4. I guess in line 225 a word is missing (The ... corresponding to).  

Reviewer 4 Report

The article is interesting. I suggest few corrections:

Introduction - line 56 - "for electrochromism and capacitor applications" - point mark in the end

Experimental section -line 84 -  "was prepared by Zheng et al had reported" - could be reformulated as "was prepared using method reported by Zheng et al"

MoO3 nanorod coated film for electrochromic measurement was deposited on indium tin oxide (ITO) coated glass? If yes, maybe it will be useful to add this remark.

Line 97 - "PVDF (polyvinylidenefluoride).A" - a space missing

To be more clear, time and temperature used to obtain the structures presented in Fig 1 and 2 can be mentioned

Line 116 - The phrase "Morphology characterization was conducted using SEM ...." - can be another paragraph.

Line 118 - "as shown in Fig. 2 b" must be replaced with "as seen in Fig. 2a inset"

Figure 3 caption must be corrected

In Figure 3 caption, it is mentioned at "170 oC" and in the image 3 e is written 120 oC - please clarify

Line 157 -  "... these nanoparticles aggregate to form nanoparticles ..." - it must be reformulated

Figure 4 caption "Cyclic voltammograms of MoO3 nanorod electrodes at a scanning rate of 50. mV/s. The transmittance changes of (a) nanorods ..." can be changed with "Cyclic voltammograms of MoO3 nanostructured electrodes at a scanning rate of 50 mV/s. The transmittance changes of (a) nanorods ...

Lines 208 - 210 - "the electrochemical performance of MoO3 nanorods was evaluated by three different electrode structures with 0.5 m Na2SO4 as electrolyte. In the following experiments, we also compared the capacitors experiments with MoO3 particles and MoO3 nanorod clusters." - in my opinion can be changed with "the electrochemical performance with 0.5 m Na2SO4 as electrolyte. We compared the MoO3 nanorods capacitors performance with MoO3 particles and MoO3 nanorod clusters."

From Fig. 7 caption, the description for images e and f is missing

Line 233 - "different current densities of 0.5, 1, 3, 5, 8 A/g as shown in Fig. 7 b" - the text is not corresponding with the graph legend presented in image 7 b

Line 255 - "supercapacitor. . However" - correction needed

Conclusions line 266 -  "The average radius of the synthesized nanorods is about 100 nm." - in the SEM images presented in the manuscript, the average radius is not measured; use a specific software to measure the dimensions

Round 2

Reviewer 1 Report

The authors have edited the manuscript to take into account most of my previous comments.   The English is still an issue. 

Reviewer 3 Report

ok

Author Response

Thank you very much for your valuable suggestions on our manuscript.

Reviewer 4 Report

The manuscript was improved and can be accepted for publication.

Author Response

(The authors gave the same response as above.)
